# Beyond Myelin Oligodendrocyte Glycoprotein and Aquaporin-4 Antibodies: Alternative Causes of Optic Neuritis

**DOI:** 10.3390/ijms242115986

**Published:** 2023-11-05

**Authors:** Giacomo Greco, Elena Colombo, Matteo Gastaldi, Lara Ahmad, Eleonora Tavazzi, Roberto Bergamaschi, Eleonora Rigoni

**Affiliations:** 1Multiple Sclerosis Centre, IRCCS Mondino Foundation, 27100 Pavia, Italy; giacomo.greco@mondino.it (G.G.); elena.colombo@mondino.it (E.C.); lara.ahmad@mondino.it (L.A.); eleonora.tavazzi@mondino.it (E.T.); roberto.bergamaschi@mondino.it (R.B.); 2Department of Brain and Behavioral Sciences, University of Pavia, 27100 Pavia, Italy; 3Neuroimmunology Research Unit, IRCCS Mondino Foundation, 27100 Pavia, Italy; matteo.gastaldi@mondino.it

**Keywords:** neuro-ophthalmology, optic neuritis, visual loss, AQP4, MOG, GFAP, CRION, sarcoidosis, optic perineuritis

## Abstract

Optic neuritis (ON) is the most common cause of vision loss in young adults. It manifests as acute or subacute vision loss, often accompanied by retrobulbar discomfort or pain during eye movements. Typical ON is associated with Multiple Sclerosis (MS) and is generally mild and steroid-responsive. Atypical forms are characterized by unusual features, such as prominent optic disc edema, poor treatment response, and bilateral involvement, and they are often associated with autoantibodies against aquaporin-4 (AQP4) or Myelin Oligodendrocyte Glycoprotein (MOG). However, in some cases, AQP4 and MOG antibodies will return as negative, plunging the clinician into a diagnostic conundrum. AQP4- and MOG-seronegative ON warrants a broad differential diagnosis, including autoantibody-associated, granulomatous, and systemic disorders. These rare forms need to be identified promptly, as their management and prognosis are greatly different. The aim of this review is to describe the possible rarer etiologies of non-MS-related and AQP4- and MOG-IgG-seronegative inflammatory ON and discuss their diagnoses and treatments.

## 1. Introduction

Subacute vision loss associated with retrorbital pain that worsens during eye movements are clinical hallmarks of optic neuritis (ON) [1]. This term encompasses a vast group of inflammatory disorders of the optic nerve, with many possible causes. A first important distinction is among typical and atypical ON. Typical ON is characterized by unilateral, mild to moderate vision loss with dyschromatopsia and pain during eye movement. It is generally responsive to corticosteroid treatment, with short, segmental lesions in MRI imaging; optic disc edema is found in under a third of these cases [2,3,4]. Typical ON is either idiopathic or a manifestation of Multiple Sclerosis (MS), as half of MS patients will develop ON during their disease course [5]. On the other hand, atypical features, such as prominent optic disc edema, poor treatment response, and bilateral involvement are often associated with autoantibodies against aquaporin-4 (AQP4) and Myelin Oligodendrocyte Glycoprotein (MOG). More specifically, Neuromyelitis Optica Spectrum Disorder (NMOSD)-related ON presents with a severe visual acuity loss, a limited response to acute-phase treatments, and poor recovery [6]. Bilateral involvement is seen in around 20% of patients, much more commonly than MS-ON [7]. Optic nerve MRI lesions tend to be longitudinally extensive, and chiasmal involvement is peculiar, found in up to two thirds of cases [8]. On the other hand, while MOG-antibody-associated disorder (MOGAD)-related ON can also present with a profound visual acuity drop, it is often greatly responsive to steroid treatment [9]. Optic disc edema is found in 80% of patients, and it is sometimes associated with peripapillary hemorrhages [8]. Pain is a frequent feature, and 50% of patients have bilateral involvement [10,11]. In cases of non-MS-related ON, the first crucial diagnostic step for the clinician is to request AQP4 and MOG antibodies, possibly with dependable laboratory assays [7,8]. If both antibodies return negative, a wide spectrum of differential diagnoses should be considered. The scope of this paper is to review and describe the rarer etiologies of inflammatory ON that are distinct from MS and the above-mentioned antibody-related disorders.

## 2. Optic Neuritis Definition and Classification

The first diagnostic criteria for ON have recently been formulated by a panel of international experts. They employ a combination of clinical features and paraclinical tests to help confirm the diagnosis of ON [12]. The clinical criteria include a monocular, subacute loss of vision associated with orbital pain that worsens during eye movements, reduced contrast and color vision, and relative afferent pupillary deficit (RAPD). In addition, at least one positive paraclinical test is required, such as Optical Coherence Tomography (OCT) findings, MRI (contrast-enhancing or high signal in the optic nerve), and biomarkers such as serum antibodies to AQP4 or MOG, or CSF oligoclonal bands [12].

Moreover, this position paper also proposed a new classification of ON, based on different anatomical compartments that can be affected by the inflammatory process. The classification is complex but relevant to this paper for mainly two reasons. First, retinal degeneration due to paraneoplastic antibodies, which generally presents as retinopathy with visual loss and signs of vitreal inflammation without retrobulbar optic nerve involvement, is now included as a cause of optic neuritis involving the so-called prelaminar compartment (i.e., the most anterior optic nerve section, running from the lamina cribrosa to the retina). Secondly, the orbital compartment, including the retrobulbar portion of the optic nerve, its sheath, and the surrounding orbital tissues, is described as another possible localization of ON. Here, pain is a prominent feature, as the optic nerve has a dural sheath, and the inflammation can spread to (or originate from) the perineural tissues, defining an entity called perineuritis. Of note, perineuritis can often overlap with optic neuritis; it could be argued that as soon as a patient complains of vision loss besides pain, the inflammatory process has involved the optic nerve itself, and both entities coexist. With this anatomical classification, both entities can be defined as optic neuritis, affecting the orbital compartment.

The other anatomical compartments include pathological entities that can involve the whole CNS such as MS, MOG-antibody-associated disorder (MOGAD), and Neuromyelitis Optic Spectrum Disorders (NMOSDs) (i.e., the “brain” compartment) and systemic diseases, which can affect other organs outside of the CNS (the “whole body” compartment).

Once the suspicion of ON is established, the classical phenotype is that of typical optic neuritis. On the other hand, some patients present with atypical features, including severe vision loss, bilateral involvement, poor steroid responsiveness or steroid dependence, severe optic disc edema, and childhood or late adult onset [13]. This clinical picture is labeled as atypical optic neuritis and should prompt serum testing for AQP4 and MOG antibodies [13]. When those two antibodies return negative, a broad differential spectrum opens, which includes other forms of optic neuritis—which are either associated with different autoantibodies, a paraneoplastic process, or other systemic conditions. 

## 3. Antibody-Associated Optic Neuritis

### 3.1. Glial Fibrillary Acidic Protein (GFAP) Antibody-Associated Optic Neuritis

Glial fibrillary acidic protein (GFAP) antibody-associated astrocytopathy is a recently described inflammatory condition presenting as meningoencephalitis with various possible signs of CNS involvement, including movement disorders, seizures, brainstem attacks, and myelitis [14,15,16]. A brain MRI can show a characteristic radial perivascular enhancement, although a leptomeningeal enhancement pattern has also been described [17,18]. Most patients have an inflammatory CSF, with prominent pleocytosis [14]. The diagnosis is made by testing for GFAP-IgG in the CSF, as positivity in the serum can only be non-specific [19]. Most patients respond well to acute treatment with intravenous steroids, although some cases may be refractory and may require plasma exchange and intravenous immunoglobulins, with mixed results [14]. The disease is monophasic in up to 89% of patients, and there is early evidence showing that visual symptoms are associated with a relapsing disease course and might help to identify patients who require chronic immunosuppression [20]. 

The visual system is frequently affected (25% of patients), and the most common symptoms are blurred vision or transient visual obscuration; however, central visual acuity is often preserved [14,20]. The most peculiar neuro-ophthalmological sign is bilateral optic disc edema, which can easily be overlooked as it is asymptomatic in more than half of patients [20]. The etiology of optic disc edema is unclear. The mildness of symptoms and bilateral involvement resemble papilledema, but the majority of patients have normal opening pressure, with few exceptions [14,21]. Studies on animal models have shown the importance of GFAP for blood–brain barrier integrity [22], and human fluorescein angiography studies in GFAP astrocytopathy patients showed prominent venular leakage [23]. These findings, coupled with the radial, perivenular enhancement of MRI, suggest that the source of inflammation in GFAP astrocytopathy could be a primary venular or perivenular inflammatory process.

ON is a rare visual manifestation in GFAP astrocytopathy, accounting for 24% of patients with visual involvement [20]. However, the cases reported in the literature often lack sufficient clinical information to assess the Petzold criteria, and it is possible that some patients were erroneously classified. GFAP antibodies have also been associated with another, rarer phenotype: bilateral optic neuritis with severe visual impairment and a limited response to acute-phase treatment including high doses of intravenous steroids and plasma exchange [20,24,25,26,27]. This appears to have an axonal pattern of damage according to the OCT and Visual Evoked Potential (VEP) findings, hence the resemblance to AQP4 optic neuritis [20].

### 3.2. Collapsin Response Mediator Protein 5 (CRMP5) Antibodies Associated with Optic Neuritis

Antibodies against collapsin response mediator protein 5 (CRMP5-IgG) have been identified as markers of a wide range of neurological paraneoplastic autoimmunities, presenting as myelopathy, cranial neuropathies, polyneuropathies with painless mixed axonal-demyelinating patterns, and basal ganglionitis [28]. The hallmark neuro-ophthalmic manifestation of CRMP5 autoimmunity is a complex phenotype of optic neuritis and retinitis with vitreous inflammatory cells, and it is almost invariably associated with optic disc edema [29]. A typical patient is a middle- or older-aged person who presents with a painless bilateral or sequential subacute vision loss of mild to moderate entity. The median High-Contrast Visual Acuity (HCVA) at presentation is 20/40, although it can range from 20/20 vision to counting fingers vision. Orbit MRI is often unremarkable and lacks any pattern of contrast enhancement [30]. Fluorescein angiography shows optic nerve head hyperfluorescence and occasionally shows subretinal edema. An electroretinogram (ERG) is often abnormal due to the inflammatory retinal involvement. Rare autoptic reports show CD8 T lymphocytes infiltrating the optic nerve and spinal cord [31].

The two tumors that are the most associated with CRMP5-IgG are small-cell lung cancer (SCLC) and, less commonly, thymoma. Crucially, up to two-thirds of patients develop this paraneoplastic syndrome before the diagnosis of the underlying neoplasm [28,29,30,31]. It is therefore mandatory to screen thoroughly for occult tumors upon diagnosis, especially SCLC and mediastinal masses. It can be challenging to differentiate CRMP-IgG-related optic neuropathy from a cancer-related leptomeningeal invasion of the optic nerves or radiation-induced damage on orbital MRI studies, as these can show various patterns of perineural enhancement [32]. It is therefore advisable that adult patients with subacute painless loss of vision, optic disc edema, and posterior vitritis should be tested for CRMP-5-IgG in the CSF. Treatment of the underlying tumor is paramount, and adjunctive therapy with intravenous steroids, plasma exchange, and immunoglobulin can be considered [33]. Around half of the patients show an improvement in visual function; the overall prognosis is related to the primary tumor [28].

### 3.3. Other Autoantibodies

Besides CRMP5 and GFAP, more antibody-mediated autoimmune optic nerve involvements have been proposed by various reports, often in the context of paraneoplastic autoimmunity. One of the earliest to be discovered was the presence of recoverin-IgG antibodies. Recoverin is a protein that is predominantly found in retinal photoreceptor cells, and a 1993 report was the first to find antibodies against recoverin in a middle-aged patient with oat cell lung carcinoma [34]. Subsequent reports showed that recoverin antibodies are present in a subset of patients with cancer-associated retinopathy [35], which has a slow, progressive course and can cause very severe visual loss. As with many paraneoplastic manifestations, the prognosis is dependent on the treatment of the underlying tumor. 

Antibodies against AQP1 target a different channel in the aquaporin family than AQP4, and they were investigated as markers of autoimmunity patients with suspected NMOSD [36]. They appear to be specific for inflammatory CNS disorders, as they are not found in healthy controls [36,37]. AQP1 antibodies are coexistent with AQP4 antibodies in 33% of cases, and their titres are strongly correlated [36]. AQP1 positive patients present with predominant spinal involvement, and a study found that patients with AQP1-IgG (with or without AQP4-IgG) presented with ON [37]. A large, more recent study failed to confirm the presence of those antibodies in NMOSD patients [38]. 

Other proposed antibodies include those against SOX2, which was found in a subset of people with relapsing, seronegative optic neuritis and confirmed with indirect immunohistochemistry [39], and glycine receptor α1 subunit (GlyR), which was first found incidentally in patients with stiff-person syndrome and visual manifestations [40], and was later identified in a large, retrospective case series of ON patients [41]. These are rare phenotypes, and the precise characterization of clinical–immunological correlations is still lacking. 

## 4. Optic Neuritis Associated with Systemic Conditions

### 4.1. Sarcoidosis-Associated Optic Neuritis

Sarcoidosis is a multisystem disorder of unknown etiology, characterized pathologically by noncaseating epithelioid cell granulomas [42]. The main affected organ is the lung, which is involved in over 90% of cases, but granulomas can be found in other organs such as the heart, skin, skeletal muscle, liver, kidneys, and the nervous system, both central and peripheral [43]. Black subjects are the most affected by sarcoidosis, with a strong female predominance [44]. Disease onset is commonly characterized by systemic symptoms such as fever, night sweats, weight loss, pulmonary symptoms, lymphadenopathy, and diarrhea; pulmonary hilar lymphadenopathy is a frequent finding in chest X-rays [43]. Rarely, the disease can present with isolated extrapulmonary localizations [43]. Nervous system involvement happens in around 5–10% of cases [45]. The most common manifestation is seventh nerve palsy, followed by other cranial neuropathies, including optic neuritis, aseptic meningitis, and other neuropathies [46]. Neuro-ophthalmological manifestations are protean and range from purely ophthalmic entities—particularly anterior uveitis, but also retinal vasculitis, vitreous infiltrates, and choroidal lesions—to optic nerve involvement [47]. Subacute optic neuritis is the most frequent phenotype. Patients complain of a subacute to rapidly progressive visual loss, which is generally reported as less painful than typical ON. The degree of vision loss is severe: nadir HCVA of no light perception (NLP) should raise the suspicion of sarcoidosis-associated ON in patients with these features [45,46]. The optic disc mostly appears swollen or normal if the inflammation is retrobulbar [45]. A third of patients with sarcoidosis-associated ON have concurrent signs of ocular inflammation, such as anterior uveitis, panuveitis, vitritis, nerve fiber layer infarcts, macular exudates, and retinal vasculitis, accompanied by peculiar perivascular infiltrates known as “candle wax droppings” [43]. Sarcoidosis diagnosis is made with the pathologic evidence of noncaseating granulomas on bioptic tissue, which can be identified through whole-body imaging, such as a chest CT or fludeoxyglucose positron emission tomography (FDG-PET) [48]. Serum Angiotensin-Converting Enzyme (ACE) is elevated in 50% of patients at the time of diagnosis. CSF studies are often unremarkable in ON patients, and there are no oligoclonal bands on Isoelectric Focusing (IEF) [43]. 

Steroid responsiveness can be impressive, and sometimes, high doses of steroids are necessary to prevent relapses and worsening [49]. Long-term immunosuppressive therapy is usually needed to spare steroid use [46]. Nonetheless, some patients will develop a progressive, unrelenting course with worsening symptoms and no steroid response [49]. Optic nerve involvement due to sarcoidosis should be suspected in all patients with a slowly developing visual loss, which is only slightly painful and initially steroid responsive, especially when associated with signs of ocular inflammation in the anterior chamber. Sarcoidosis can also present with perineuritis and orbital inflammation (see below) or optic nerve head granulomas.

### 4.2. Optic Neuritis Associated with Systemic Autoimmune Disorders

Acute optic neuritis can rarely manifest in the context of systemic immune disorders. On occasion, it may even present as the initial symptom, signaling the presence of an underlying rheumatological disorder.

Isolated optic neuritis is a rare manifestation of systemic lupus erythematosus (SLE) [43]. It is strongly advised to check for AQP4 antibodies in SLE patients who develop a severe, uni-, or bilateral subacute loss of vision with pain and a limited steroid response in the acute phase, as AQP4-IgG seropositivity is common in patients with SLE who present with NMOSD, longitudinally extensive transverse myelitis, or ON [50]. It is equally advisable to treat those patients aggressively, similarly to those with prototypical AQP4 optic neuritis [51]. 

CNS involvement in Sjogren’s syndrome is infrequent (approximately 5%), and optic neuritis is present in a small percentage (4%) of these cases [52]. Similarly to SLE ON patients, the visual loss course closely resembles that of NMOSD, and AQP4 antibodies should be tested [52].

Another rheumatological disorder that can rarely present with optic neuritis is Behcet’s syndrome, an autoinflammatory disease that features recurrent orogenital ulceration, skin lesions, arthropathy, colitis, and venous thrombosis [53]. Common ophthalmic complications are anterior uveitis with hypopyon and vaso-occlusive retinal vasculitis [54,55]. A retrospective case series showed that patients present with a subacute, progressive loss of vision with optic disc edema and minimal retrorbital pain [56]. The administration of corticosteroids improves symptoms, and two-thirds of patients return to their normal vision [56]. Adjunctive immunosuppressive therapy is only needed because of the primary systemic disorder.

Granulomatosis with polyangiitis (GPA, formerly known as Wegener granulomatosis) is a small-medium vessel necrotizing vasculitis, which is a component of a vast spectrum of disorders called anti-neutrophil cytoplasmic antibody (ANCA)-associated vasculitides (AAV) [52]. It involves the upper and lower respiratory tract, including the sinonasal cavities, the kidneys, skin, skeletal muscle, and nervous system [52]. Around 30% of patients with GPA are reported to have ocular involvement, including scleritis, conjunctivitis, and peripheral ulcerative keratitis [52]. Both retrobulbar optic neuritis and optic perineuritis have been reported; these present with severe visual loss, which is sometimes responsive to corticosteroids and immunosuppression [52].

Optic perineuritis (OPN) has been reported in disorders such as Behçet’s disease, Crohn’s disease, sarcoidosis, and granulomatosis with polyangiitis (GPA) [57,58,59,60,61]. Although it mostly causes orbital inflammatory syndromes with no vision loss, IgG4 -related disease can rarely present as OPN [62]. In rare cases, OPN was the presenting symptom that led to the diagnosis of the underlying disease, particularly with Behçet’s disease [60]. The clinical and radiological characteristics of secondary OPN are similar to those of idiopathic OPN, although the associated clinical features of the systemic condition help with diagnosis, e.g., the presence of anti-neutrophil cytoplasmic antibodies (ANCAs) for GPA diagnosis [63]. Treatment involves the use of similar corticosteroid regimens, with an optimal response; immunosuppressive treatments are employed more frequently due to the presence of the underlying systemic disorder [57]. 

## 5. Other Idiopathic Optic Neuritis

### 5.1. Chronic Relapsing Inflammatory Optic Neuropathy (CRION)

Chronic relapsing inflammatory optic neuropathy (CRION) was described systematically for the first time by Desmond Kidd et al. in a seminal report in 2003 [64]. It is a rare form of inflammatory optic neuritis with peculiar characteristics. Patients present with a unilateral or bilateral subacute loss of vision, where pain is a prominent feature [65,66]. Pain is severe and persistent, and it can precede visual loss by a few days, worsening upon the visual deficit onset [65,66]. The visual loss is severe, as two-thirds of patients reach an HCVA at nadir of 20/200, compared with 30–35% in typical optic neuritis [64]. The pathological process is demyelinating in nature, as indicated by the VEP findings, showing latency prolongation without amplitude reduction or conduction blocks in more severe cases. ERG is normal, consistent with a pure optic nerve disorder. The response to an acute, high-dose steroid treatment is dramatic: visual acuity improves substantially, and the pain eventually subsides, although it can prove to be refractory [64,65]. Another defining feature of CRION is its relapsing nature and striking steroid dependency: a prolonged oral steroid course is mandatory after the acute event to help complete the healing process and prevent further relapses [64]. CRION can closely resemble the clinical course and steroid responsiveness of other ON types, particularly sarcoidosis-related optic neuropathy, and is therefore a diagnosis of exclusion. It is necessary to look for systemic sarcoidosis, whose signs can be made apparent after many years. CSF studies, including IEF, should be performed to exclude the presence of oligoclonal bands. With the advent of more specific and accurate MOG-IgG assays, those antibodies were found in a significant proportion (up to 66%) of CRION patients, who were therefore diagnosed with MOG-antibody-associated disorder (MOGAD) [67]. A set of diagnostic criteria has been proposed to identify CRION patients based on five cardinal features that combine clinical, laboratory, and imaging features, history, and the response to treatment [66].

No official guidelines on the treatment of CRION exist, but the current practice is to treat relapses similarly to other inflammatory disorders such as MS—high-dose intravenous Methylprednisolone for 3–5 days, and Plasma Exchange if needed. It is crucial to follow acute-phase therapy with chronic oral prednisolone therapy, starting with 1 mg/kg followed by a gradual tapering [64]. It is important to note that although the response to acute corticosteroid treatment allows for a near restitutio ad integrum in initial attacks, subsequent relapses dent the optic nerve axonal fiber layer, and cumulative damage leads to structural changes in the retinal nerve fiber layer (RNFL) and ganglion cell layer (GCL) complexes, with consequent poor visual function. Also, CRION patients have a three-fold higher risk of a relapsing course compared with idiopathic optic neuritis [65]. Therefore, patients should be carefully followed-up with in time to promptly identify the need for chronic immunosuppression with agents such as azathioprine, mycophenolate mofetil, and methotrexate [68]. Still, some evidence points to the existence of interattack progression, likely as a result of such a severe inflammatory process to involve the axonal component as well as the myelin sheath [66].

### 5.2. Primary Idiopathic Optic Perineuritis

The term optic perineuritis (OPN) indicates an optic neuropathy caused by the inflammation of the optic nerve sheath [52]. Clinically, OPN can be difficult to distinguish from ON, as it is characterized by an acute to subacute unilateral visual loss, pain during eye movements, and a normal or swollen optic disc [69]. Dyschromatopsia, visual field abnormalities, and RAPD can all be present [69]. Pain is a prominent feature, as the optic nerve sheath contains pain fibers of trigeminal origin [70]. OPN may be a primary syndrome with no apparent underlying cause (idiopathic OPN), or it may part of an identifiable systemic disorder (secondary OPN) [63]. OPN is also a recognized feature of MOGAD, and the presence of serum MOG-IgG easily confirms the diagnosis; this will not be discussed here [71].

OPN is primary, or idiopathic, in most cases [63]. It is a rare disorder, and most of the information on this disease entity comes from a large case series collected through literature reviews. OPN is frequently misdiagnosed as ON, and some features are indeed overlapping. In both OPN and ON, females are slightly more affected than males, and the clinical presentation is similar, with unilateral, subacute vision loss accompanied by pain during eye movement [72]. Optic disc edema is frequent in OPN, as it is seen in around two-thirds of patients [63,72]. Patients with primary OPN tend to be slightly older (mean age of onset is 40–41 years) [63]. Also, the course of vision loss is more subacute in primary OPN, developing over weeks rather than days, and patients may present with eye pain without much of a visual impairment. HCVA loss is mild to moderate in OPN, with more than half of patients having 20/20 vision at diagnosis [63,72]. Consistent with central vision sparing, the dyschromatopsia is milder than what is typically seen in patients with ON [63,72]. Eye involvement is mostly unilateral, and exceptions are rare. Visual field alterations include arcuate defects, central and paracentral scotomas, and peripheral defects; sometimes there are no abnormalities at all [73]. Mild signs of orbital involvement, such as ptosis and ocular movement limitations, are infrequent but helpful for diagnosis, as they are not associated with ON [63,72].

Differential diagnosis for OPN is wide and includes malignant entities, such as optic nerve sheath meningioma, orbital pseudotumor, sarcoidosis, lymphoma, perioptic hemorrhage, and Erdheim–Chester disease [74]. Orbital MRI studies are of great help, especially gadolinium-enhanced fat-saturated T1-weighted MRI. A hallmark finding is a peculiar circumferential optic nerve sheath enhancement (“tramtrack” on axial views and “doughnut” on coronal views) [75]; MRI scans can also show a “streaky” enhancement of periorbital fat [76]. Radiological evidence of the inflammatory process involving the optic nerve itself is rare but has been reported, differently to ON patients, where enhancement and T2 hyperintensities are located exclusively in the optic nerve itself [75]. A close and careful examination may show a subtle enhancement of the extraocular muscles and/or sclera in addition to the characteristic changes in the optic nerve sheath. Calcifications could be a hint for optic nerve sheath meningioma, another possible etiology of the tramtrack sign, and CT studies may be helpful to rule out this diagnosis [77]. As for the other possible causes, it is advisable to look for signs of other organs’ involvement, and an expanded evaluation for granulomatous disease, including testing for antineutrophil cytoplasmic antibodies (perinuclear–antineutrophil cytoplasmic antibody and cytoplasmic–antineutrophil cytoplasmic antibody, p-ANCA and c-ANCA) and chest radiography. T. pallidum serology is also useful [63,69].

The response to corticosteroid therapy is dramatic and helps to make a diagnosis. Treatment with 1 mg/kg prednisolone produces striking eye pain relief, often within the first 24 h, accompanied by a quick improvement in vision [72]. On the other hand, relapses are frequent in steroid withdrawal, and tapering should be conducted with caution, as patients will continue to lose vision, in some cases irreversibly, without proper treatment [63,72]. The prognosis for visual loss is optimal, although the main determinant is how quickly the disease is diagnosed; delays of more than a month could prove to be costly.

Table 1 summarizes the clinical and instrumental features of the main types of seronegative non-MS-related inflammatory ON.

## 6. Other Causes of Optic Neuritis

Other causes of optic neuropathies exist. Bacterial agents, such as T. pallidum, B. henselae, B. burgdorferi, and viral agents (herpes simplex virus, varicella zoster virus, and cytomegalovirus) have been associated with acute optic nerve involvement [84]. Systemic features, including fever, meningitis, other cranial nerve palsies, and encephalitis, are often coexistent [84]. Similarly, other etiologies of optic nerve dysfunction such as compressive or drug-related optic nerve dysfunction present with diverse histories and clinical features. Imaging of the brain and optic nerve is advisable to exclude compressive causes, as well as a thorough review of medications and possible toxic exposures. These cases enter a diagnostic spectrum of their own, for which differential diagnosis is large and outside the scope of this review.

## 7. Conclusions

ON is a frequent occurrence in neurological practice. It is a heterogenous group of disorders that share a common pathological ground of optic nerve inflammation and the key clinical features of vision loss and retro orbital pain. These can be more or less prominent and accompanied by different signs and symptoms, which we discussed here. Typical optic neuritis is the most common presentation, especially in young adults; when atypical features are present, such as bilateral involvement, prominent optic disc edema with hemorrhages, longitudinally extended lesions on orbit MRI, and poor steroid response, it is wise to ask for serum antibodies to AQP4 and MOG, which are commonly associated with atypical ON. When those antibodies are negative, rarer causes should be considered. This review is meant to assist clinicians in those rare cases of seronegative optic neuritis to provide guidance on possible diagnostic opportunities and request appropriate testing. We particularly encourage clinicians to look for accompanying symptoms and signs that could guide the diagnosis, such as bilateral optic disc edema for GFAP-antibody associated ON, concomitant retinitis, vitreal inflammation and a lack of MRI findings in CRMP5-IgG ON, the dramatic response and dependency on steroid treatment for CRION, the associated uveitis and anterior chamber inflammation for sarcoidosis-related ON, or the prominent pain and central vision sparing of OPN. We summarize our description with a diagnostic flowchart in Figure 1. Correct and expedite diagnosis is vital to start and maintain appropriate treatment in all of these conditions, as prognostic implications are relevant. Nonetheless, some cases of seronegative ON remain idiopathic; future studies aimed at elucidating new autoantibody reactivities will help to define new pathological entities with distinctive prognostic features. New diagnostic tests could then become available in clinical practice to promptly identify patients with new autoantibodies and specific features to undertake the right therapeutic management and improve outcomes.

## Figures and Tables

**Figure 1 ijms-24-15986-f001:**
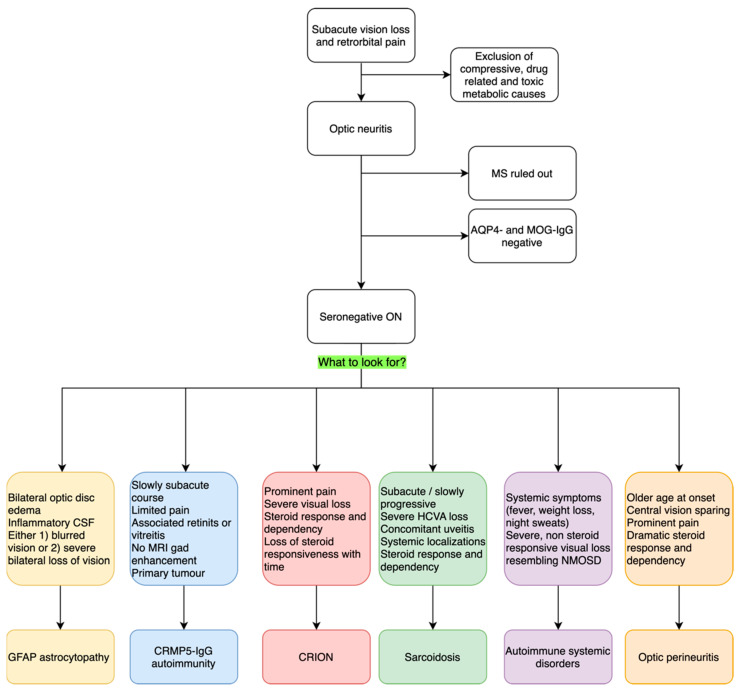
Proposed diagnostic flowchart for seronegative inflammatory ON. ON: optic neuritis; MS: Multiple Sclerosis. AQP4: Aquaporin-4; MOG: Myelin Oligodendrocyte Glycoprotein; Gd: Gadolinium; HCVA: High-Contrast Visual Acuity; NMOSD: Neuromyelitis Optic Spectrum Disorder.

**Table 1 ijms-24-15986-t001:** Clinical and radiological features of the main types of seronegative non-MS-related ON. HCVA: High-Contrast Visual Acuity. NLP: no light perception. Gd: gadolinium. IPL: inner plexiform layer. GCL: ganglion cell layer. pRNFL: peripapillary retinal nerve fiber layer.

Clinical Features	GFAP-IgG	CRMP5-IgG	CRION	Sarcoidosis	Optic Perineuritis
Sex distribution	F:M 1:1 [15]	F:M 1:1 [28]	F:M 2:1 [66]	F:M 1:1 [78]	2.5:1 [63]
Age at onset (median/mean)	44–50 years	69 years	35.7 years	42–48 years	41 years
Retro-orbital pain	Rare	Rare	Prominent	Frequent	Prominent
Visual loss severity	(1) No HCVA loss (bilateral optic disc edema) or (2) severe bilateral vision loss	Variable, median around 20/40	Severe: 20/200 in two-thirds of cases	Severe; often NLP	Mild, often central vision sparing
Optic disc edema	Very frequent; may be asymptomatic	Very frequent	Variable	Variable	Frequent
Visual loss course	Subacute	Subacute	Subacute	Subacute to slowly progressive	Subacute to slowly progressive
MRI findings	Symmetrical FLAIR hyperintensities involving basal ganglia, thalami, internal, and external capsules; Linea radial (or leptomeningeal) contrast enhancement	Optic nerve may be normal; rarely T2 hyperintensities with no Gd enhancement; brain MRI can show basal ganglia, medial temporal lobe, extensive white matter, hippocampus, cerebellum, insula, thalamus, and frontal lobe T2 hyperintensities	Normal in 40% of patients; possibly isolated T2 hyperintense lesions in periventricular or juxtacortical white matter	Leptomeningeal or pachymeningeal enhancement, including cranial nerve enhancement, MS-like white matter lesions, optic nerve T2 hyperintensities, focal parenchymal areas of contrast enhancement	Ill-defined, circumferential optic nerve sheath enhancement (“tramtrack” or “doughnut” sign) to be differentiated from optic nerve sheath meningioma, orbital pseudotumor, or sarcoidosis; possible enhancement of orbital fat surrounding optic nerve sheath or extraocular muscle
OCT findings	In cases of optic disc edema: normal outer retina with elevated retinal nerve fiber layer thickness [20]	Increased RNFL consistent with optic disc edema (acute phase); atrophy of the outer retinal layers was noted, with deepening of the foveal depression; hyperreflective dots (atrophic stage) [79,80]	Severe thinning in RNFL and thinning in intra-retinal segments of IPL, GCL, RNFL, and TMV compared with NMOSD and MS-related ON [81]	Subretinal fluid, macular edema, and loss of retinal architecture; no alterations in patients without clinical optic involvement [82]	Significant thinning of average peripapillary RNFL [83]
Steroid dependency	Limited	None	Severe	Severe	Severe

## Data Availability

Not applicable.

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
