# Peer review of "Beyond Myelin Oligodendrocyte Glycoprotein and Aquaporin-4 Antibodies: Alternative Causes of Optic Neuritis"

_ijms, 2023, doi:10.3390/ijms242115986_

Round 1

Reviewer 1 Report

Comments and Suggestions for Authors

In this review the authors discussed possible and rare etiologies of optic neuritis when AQP4 and MOG antibodies are negative, and then discussed their diagnosis and treatment. The topic of this review is interesting. Some concerns and suggestions are listed as below:

In the abstract, why do you mean by saying 'typical optic neuritis is associated with multiple sclerosis'? This is not always true in clinical practice. Multiple sclerosis is thought to be relatively uncommon in the Asia Pacific region with prevalence estimated between 0 and 20 per 100,000.

The authors mentioned that typical optic neuritis represents the vast majority of presentations, and is characterized by a unilateral, mild to moderate vision loss. However, blindness can be noted (one or both eyes) in patients with NMOSD.

When those two antibodies return negative, a broad differential spectrum opens. How long will it take for testing serum AQP4 and MOG antibodies in your center (Pavia)? How will you deal with optic neuritis patients if antibody results will not be available in the following days? Which methods should be used? Some may argue that some methods can cause false negative for these antibodies.

The English of this manuscript should be edited. For example, 'Optic neuritis (ON)' should be 'ON' in line 364.

Bacterial infections, including Lyme disease, cat-scratch fever and syphilis, or viruses, such as measles, mumps and herpes, can cause optic neuritis.

Some drugs and toxins have been associated with the development of optic neuritis. 

In Table 2 (Clinical and radiological features of the main types of optic neuritis), potential risk factors (such as sex, age or race) and OCT (OCT-A) biomarkers of each disease should be discussed in details. 

How about AQP1-antibodies?

Comments on the Quality of English Language

fine

Reviewer 2 Report

Comments and Suggestions for Authors

This article addresses a clinically very relevant topic, as the etiological attribution of optic neuritis is not very straightforward. Nevertheless, there are some aspects that should be added:

  1. - For the review, it would be important to at least mention the findings in classical MS, NMOSD, and MOGAD in the table.
  2. - It would be beneficial if the MRI findings, in particular, are described more clearly in the table to make differences noticeable. Visual evoked potential (VEP) results and, especially, OCT studies and reports should be systematically included here.
  3. - CRION: Please also discuss the use of B-cell depleting therapies as a treatment option.
  4. - A certain diagnostic guideline is missing. When should the antibodies be tested? Should everything be tested in cases of optic neuritis? A figure would be helpful. Minor findings:
  5. - Page 1, Line 26: There is a missing space. - Please consistently use abbreviations (e.g., ON) and especially spell them out when first used (e.g., CSF).

Reviewer 3 Report

Comments and Suggestions for Authors

This is a useful and comprehensive review of rare etiologies of optic neuritis, i.e. cases negative for autoantibodies against AQP4 and MOG. They are clasified in autoantibody associated, systemic-conditions associated and others (including perineuritis). It would be a good chapter in a book of neuro-ophthalmology.

The few problems I have found are of form and not of content.

Acronyms should be explained when they appear for the first time in the text, e.g.: IED (line 194), PLEX (line 275).

The only table in the paper is erroneously numbered as Table 2. Please correct.

Round 2

Reviewer 1 Report

Comments and Suggestions for Authors

My previous concerns have been addressed by the authors.

Comments on the Quality of English Language

none

Reviewer 2 Report

Comments and Suggestions for Authors

-